# A Mathematical Framework for Characterizing Dependency Structures in Multimodal

## Abstract

Dependency structures between modalities have been utilized explicitly and implicitly in multimodal learning to enhance classification performance, particularly when the training samples are insufficient. Recent efforts have concentrated on developing suitable dependence structures and applying them in deep neural networks, but the interplay between the training sample size and various structures has not received enough attention. To address this issue, we propose a mathematical framework that can be utilized to characterize conditional dependency structures in analytic ways. It provides an explicit description of the sample size in learning various structures in a non-asymptotic regime. Additionally, it demonstrates how task complexity and a fitness evaluation of conditional dependence structures affect the results. Furthermore, we develop an autonomous updated coefficient algorithm auto-CODES based on the theoretical framework and conduct experiments on multimodal emotion recognition tasks using the MELD and IEMOCAP datasets. The experimental results validate our theory and show the effectiveness of the proposed algorithm.

## 1 Introduction

Multimodal learning is recently an active research area in machine learning aiming at jointly extracting information and learning knowledge from different categories of data, such as images, audios, and texts (Ngiam et al., 2011; Zadeh et al., 2019; Kiela et al., 2020). In multimodal learning, a critical issue is to design efficient algorithms to extract features from various modalities, such that the label information can be effectively extracted for classification, especially when the number of training samples is insufficient to learn a huge and complex multimodal structure. There have been many kinds of literature addressing this issue with different kinds of algorithms (Baltrušaitis et al., 2018), in which the main research stream focuses on extracting features from several modalities that are both relevant to the label and connected to one another (Gao et al., 2020; Ma et al., 2020; Summaira et al., 2021). The effectiveness of such algorithms could have resulted from the intuition that the labels appear to be the common patterns shared between different modalities in many real multimodal datasets. For multimodal data with such property, designing modality features with higher correlations will implicitly force the algorithm to search for more informative features to the label, and hence can often require fewer training samples to achieve good performance.

From the statistical learning aspects, such benefits can be interpreted as that the modalities and the label follow a conditional dependency structure where modalities are independent of each other once the label is given. Since it is in a relatively low-dimensional space, it will demand less number of samples to learn a good representation (Varma et al., 2019). Thus, there can be several factors affecting the classification performance including (i) the number of labeled training samples, (ii) the fitness of the conditional dependency structures to represent the true one, and (iii) the complexity of the discrimination task.

There are works exploiting appropriate dependency structures to achieve good performances by designing effective networks, fusion approaches (Zadeh et al., 2017; Liu et al., 2018; Nagrani et al., 2021) and objective functions (Sohn et al., 2014; Sutter et al., 2020; Piergiovanni et al., 2020). However, most existing works focus on designing learning algorithms and architectures without a theoretical understanding of the training sample size in multimodal problems, which potentially limits the performance, especially for complicated multimodal problems.

In this paper, we consider the model to learn a linear combination of two types of estimators (Piergiovanni et al., 2020) representing the general dependency structure and the considered conditional dependency structure. Then, we propose a testing loss function to evaluate the performance of the trained model in a non-asymptotic regime. It considers the average performance of the linear combined estimator over a certain number of training samples. Note that it is parametrized by a linear combination coefficient of different types of estimators. Minimizing the testing loss will give us the optimal coefficient which determines the optimal estimator which can lead to more informative classification features. The optimal coefficient can also be used to characterize the conditional dependency structure in the process. The detailed mathematical formulation and interpretations are presented in Section 2 and Section 3.

In particular, we show that the explicit analytical solution of the optimal coefficient is inversely proportional to the number of labeled training samples and the fitness measurement of the conditional dependency structure. Also, it is proportional to the complexity of the learning task, which we will discuss in Section 2. For instance, when the number of samples is insufficient to learn a high-dimensional model, it will be preferable to fit the low-dimensional conditional dependency structure which requires less number of parameters. Meanwhile, the coefficient for the estimator corresponding to the conditional structure will be large. Therefore, the optimal coefficient essentially indicates the efficient model that one shall choose for predicting the label in the multimodal problem with the number of training samples taken into account. Moreover, our approach essentially provides guidance and theoretical understanding for designing efficient multimodal algorithms to utilize the sample size information and different dependency structures.

At last, we extend our theoretical results and propose an autonomous updated coefficient on dependency structures (auto-CODES) algorithm by exploiting parametric models. It can compute the weights on different dependency structures automatically with the features evolving in deep neural networks. The experiments on the emotion recognition tasks with the MELD and IEMOCAP datasets validate our theoretical results and show the effectiveness of the algorithm. The main contributions of this paper can be summarized as follows:

- We propose a novel theoretical framework for multimodal analyses to characterize the influence of the conditional dependency structure. To the best of our knowledge, it is the first work to give an explicit characterization of the number of training samples toward different dependency structures for multimodal learning in a non-asymptotic regime. Also, it quantifies the task complexity and the fitness of the conditional dependency structure, measured by the $\chi^2$-divergence, to the estimation.

- We extend the analyses from discrete to continuous data in the real world by exploiting parametric models. Furthermore, we propose an algorithm with the autonomous updated coefficient on different dependency structures (auto-CODES) based on the theoretical analyses.

- We evaluate the proposed algorithm auto-CODES on multimodal emotion recognition tasks with the widely used MELD and IEMOCAP datasets. The experimental results validate our theory and show the effectiveness of our algorithm.

Due to space limitations, the proofs of theorems are presented in the supplemental material.

## 2 PROBLEM FORMULATION AND ANALYSIS

In this section, we consider a multimodal scenario where both modalities are discrete random variables. For better illustration, we elaborate the framework in two modalities case. Specifically, we focus on the linear combination of two types of estimators. By introducing the testing loss, we evaluate the performance of the proposed estimator. Finally, we determine the optimal combining coefficient by minimizing the testing loss and illustrate the aspects that affect the optimal coefficient.

**Notation:** First, let random variables $X_1$, $X_2$ and $Y$ denote different modalities and label over finite alphabets $\mathcal{X}_1$, $\mathcal{X}_2$ and $\mathcal{Y}$, respectively. Then, $n$ sample tuples $\mathcal{D} \triangleq \{(x_1^{(i)}, x_2^{(i)}, y^{(i)})\}_{i=1}^n$ are generated in an independent, identically distributed (i.i.d.) manner from the true joint distribution $P_{X_1 X_2 Y}$, where $P_{X_1 X_2 Y}(x_1, x_2, y) > 0$ for all entries. Specifically, we consider two different estimators to approximate the joint distribution $P_{X_1 X_2 Y}$: (i) the empirical joint distribution $\hat{P}_{X_1 X_2 Y}$,

and (ii) the empirical Markov-structured distribution $\hat{P}_{X_1 X_2 Y}^{(\mathrm{M})}$, which characterize the conditional dependency structure $X_1 - Y - X_2$,

$$\hat{P}_{X_1 X_2 Y}(x_1, x_2, y) \triangleq \frac{1}{n} \sum_{i=1}^{n} \mathbb{1}\{x_1^{(i)} = x_1\} \mathbb{1}\{x_2^{(i)} = x_2\} \mathbb{1}\{y^{(i)} = y\}, \tag{1a}$$

$$\hat{P}_{X_1 X_2 Y}^{(\mathrm{M})}(x_1, x_2, y) \triangleq \hat{P}_{X_1|Y}(x_1|y)\hat{P}_{X_2|Y}(x_2|y)\hat{P}_Y(y), \tag{1b}$$

where $\mathbb{1}\{\cdot\}$ denotes the indicator function, $\hat{P}_Y$ denotes the marginal empirical distribution of label $Y$ and $P_{X_1 X_2 Y}^{(\mathrm{M})}$ denotes $P_{X_1|Y} P_{X_2|Y} P_Y$. For simplification, we consider the case when the label distribution has been learned well[1] i.e., $\hat{P}_Y(y) = P_Y(y)$, for $y \in \mathcal{Y}$.

To begin, we focus on a linearly combined estimator based on two different structures.

$$\tilde{P}_{X_1 X_2 Y} \triangleq (1 - \alpha) \cdot \hat{P}_{X_1 X_2 Y} + \alpha \cdot \hat{P}_{X_1 X_2 Y}^{(\mathrm{M})}, \tag{2}$$

where the coefficient $\alpha \in [0, 1]$ is the parameter to be designed[2].

Note that when $\alpha$ becomes zero, the proposed estimator equation 2 will degrade into a widely-used unbiased estimator $\hat{P}_{X_1 X_2 Y}$. When $\alpha$ goes to one, it will become the estimator $\hat{P}_{X_1 X_2 Y}^{(\mathrm{M})}$ representing the conditional dependency structure, i.e. $X_1 - Y - X_2$. Thus, there has an optimal combining coefficient to make the estimator equation 2 the most appropriate estimation for label prediction. In addition, the optimal coefficient can be used to characterize the dependency structure and provide theoretical insights for deriving the optimal estimation.

## 2.1 OPTIMAL COMBINATION COEFFICIENT

We define a testing loss function based on the *referenced $\chi^2$-divergence* to measure the performance of the estimator equation 2, where the referenced $\chi^2$-divergence is defined as follows[3].

**Definition 1.** *For discrete random variable $Z$ over finite alphabet $\mathcal{Z}$, and its distributions $P_Z$ and $Q_Z$, with reference distribution $R_Z$, the referenced $\chi^2$-divergence between them is defined as*

$$\chi^2_{R_Z}(P_Z, Q_Z) \triangleq \sum_{z \in \mathcal{Z}} \frac{(P_Z(z) - Q_Z(z))^2}{R_Z(z)}. \tag{3}$$

*We denote $\chi^2(P_Z, Q_Z) \triangleq \chi^2_{P_Z}(P_Z, Q_Z)$, which corresponds to the Pearson $\chi^2$-divergence.*

Based on the referenced $\chi^2$-divergence, we define the testing loss as the average divergence between the estimator (2) and the true joint distribution under the fixed training sample size $n$.

**Definition 2.** *For estimator $\tilde{P}_{X_1 X_2 Y}$ with coefficient $\alpha$ and the corresponding true distribution $P_{X_1 X_2 Y}$, the testing loss and the optimal coefficient $\alpha^*$ are defined as*

$$\tilde{\mathcal{L}}_{\mathrm{test}}(\alpha) \triangleq \mathbb{E}\left[\chi^2(P_{X_1 X_2 Y}, \tilde{P}_{X_1 X_2 Y})\right], \quad \alpha^* \triangleq \underset{\alpha \in [0,1]}{\arg\min}\, \tilde{\mathcal{L}}_{\mathrm{test}}(\alpha), \tag{4}$$

*where the expectation is taken over all $n$ i.i.d. samples generated from the true distribution.*

Then, we have the following characterization of our proposed testing loss over the linearly combined estimator and the optimal combining coefficient $\alpha^*$.

**Theorem 3.** *The testing loss as defined in equation 4 can be expressed as*

$$\tilde{\mathcal{L}}_{\mathrm{test}}(\alpha) = \left(\frac{1}{n}C + \frac{1}{n}V + \chi^2(P_{X_1 X_2 Y}, P_{X_1 X_2 Y}^{(M)})\right) \cdot \alpha^2 - \frac{2}{n}C \cdot \alpha + \frac{1}{n}(|\mathcal{X}_1||\mathcal{X}_2||\mathcal{Y}| - 1), \tag{5}$$

---

[1] Note that in many real-world datasets, such as MNIST, the label will be uniformly distributed in the training set which makes this assumption reasonable.

[2] It can be shown that the estimator equation 2 can be naturally derived from optimizing a linearly combined Log-Loss function, where we refer to the supplementary material for detailed discussion.

[3] Conventionally, such performance is computed by logarithm loss. However, in our setting, it will be ill-defined when some $(x_1, x_2, y)$ tuple is missing in training samples. By that time, we have $\tilde{P}_{X_1 X_2 Y}(x_1, x_2, y) = 0$ while $P_{X_1 X_2 Y}(x_1, x_2, y) > 0$, which would bring the logarithm loss to infinite.

*and the optimal coefficient $\alpha^*$ to minimize the testing loss equation 4 can be given as*

$$\alpha^* = \frac{\frac{1}{n}C}{\chi^2(P_{X_1 X_2 Y}, P_{X_1 X_2 Y}^{(M)}) + \frac{1}{n}C + \frac{1}{n}V}, \tag{6}$$

*where*

$$C \triangleq |\mathcal{Y}| \cdot [|\mathcal{X}_1||\mathcal{X}_2| - (|\mathcal{X}_1| + |\mathcal{X}_2|)] + 1 + a_n, \tag{7}$$

$$V \triangleq -6 \cdot \chi^2(P_{X_1 X_2 Y}, P_{X_1 X_2 Y}^{(M)}) + 2\sum_{x_2, y} \chi^2(P_{X_1|X_2 Y}, P_{X_1|Y}) + 2\sum_{x_1, y} \chi^2(P_{X_2|X_1 Y}, P_{X_2|Y})$$

$$+ |\mathcal{Y}|(|\mathcal{X}_1| + |\mathcal{X}_2|) - 2 + b_n. \tag{8}$$

*$a_n$ and $b_n$ are of the order $O(\frac{1}{n})$, which will go to constants when $n$ goes to infinity. The proof and the detailed analytical expressions of $a_n$ and $b_n$ are provided in the supplementary material.*

By considering the conditional dependency structure and tuning the coefficient $\alpha$, the improvement can be calculated by the difference in the testing losses.

**Corollary 4.** *The improvement of considering the optimal coefficient $\alpha^*$ can be given as*

$$\tilde{\mathcal{L}}_{test}(0) - \tilde{\mathcal{L}}_{test}(\alpha^*) = \frac{1}{n} \cdot \frac{C^2}{C + V + n \cdot \chi^2(P_{X_1 X_2 Y}, P_{X_1 X_2 Y}^{(M)})}, \tag{9}$$

*where parameters $C$ and $V$ are defined in Theorem 3.*

From the expressions equation 6, the optimal combining coefficient $\alpha^*$ is determined by three main factors: (i) the training sample size $n$; (ii) the fitness of the conditional dependency to describe the true one, measured by the $\chi^2$-divergence $\chi^2(P_{X_1 X_2 Y}, P_{X_1 X_2 Y}^{(M)})$ and terms in the parameter $V$; and (iii) the task complexity $C$, characterized by the number of parameters needed to estimate the joint distribution. The last characterization comes from the fact that when the task is to learn all the entries of the true distribution, the number of parameters we need corresponds to the cardinality of the sample space.

Many existing multimodal algorithms (Ma et al., 2020) focus on finding the appropriate dependency structures to approximate the true one, while the number of training samples is not sufficiently addressed. In Theorem 3, we show that the combining coefficient is inversely proportional to the number of training samples and the fitness measure of the conditional dependency to the true one. Also, it is proportional to the task complexity measured by the number of model parameters.

There are two interesting special cases for better understanding of Theorem 3 and Corollary 4. *Case 1*: When the true dependency structure is Markovian, i.e. $X_1 - Y - X_2$, the optimal coefficient will becomes $1 - V(C + V)^{-1}$, which is nearly $1 - |\mathcal{X}_1|^{-1} - |\mathcal{X}_2|^{-1}$. The cardinality terms $|\mathcal{X}_1|$ and $|\mathcal{X}_2|$ are usually large which results in that the $\alpha^*$ is quite close to 1, representing that the model should be close a Markov one [4] and the improvement is relatively large. *Case 2*: When the number of training samples is relatively small and insufficient to learn a complex model, the optimal coefficient $\alpha^*$ would be close to 1, meaning that the model behaves as a "near Markov" one and would improve from considering the conditional dependency structure. Such insights were not well captured in many multimodal algorithms, and our results essentially provide the optimal characterization of the combing coefficient among different dependency structures adjusted by the sample size and the fitness measure.

Additionally, the established expression of $\alpha^*$ can be interpreted as the optimal bias-variance trade-off (Duda et al., 1973) of the low-dimensional structure to the estimation. Note that the bias-variance trade-off in testing loss (4) is tuned by the coefficient $\alpha$ as

$$\tilde{\mathcal{L}}_{\text{test}}(\alpha) = \underbrace{\frac{1}{n}\left(C\alpha^2 + V\alpha^2 + 2C\alpha + |\mathcal{X}_1||\mathcal{X}_2||\mathcal{Y}| - 1\right)}_{\text{variance term(s)}} + \underbrace{\alpha^2 \chi^2(P_{X_1 X_2 Y}, P_{X_1 X_2 Y}^{(M)})}_{\text{bias term}}. \tag{10}$$

The variance terms will vanish with the increase in the training sample sizes. And the bias term characterizes the cost of utilizing dependency structures. Then, the coefficient $\alpha^*$ achieves the optimal bias-variance trade-off when the testing loss is minimized.

---

[4]Due to the consideration of the limited number of samples and the assumption on the distribution of label $Y$, it will not be strictly 1.

# 3 ALGORITHM WITH AUTONOMOUS UPDATED COEFFICIENT ON DEPENDENCY STRUCTURE

In this section, we build an algorithm with an autonomous updated coefficient on the dependency structures (auto-CODES) as a realization of our theoretical framework on multimodal learning. Specifically, we first extend our theory from a discrete data domain to a continuous one that can be applied to practical datasets using representations in factorization form. Then, we give the optimal coefficient $\alpha$ expressed by multimodal features and the objective loss function which linearly combines two components corresponding to different dependency structures. Finally, we give the proposed auto-CODES algorithm and the discrimination rule using maximum a posterior (MAP).

Throughout this section, we consider a dataset with $k$ modalities $X_1, \ldots, X_k$, $n$ training samples $(x_1^{(i)}, \ldots, x_2^{(i)}, y^{(i)}), i = 1, \ldots, n$, and $m$ labels. The conditional dependency structure considered here is that all the modalities are independent of each other once the label is given.

## 3.1 MULTIMODAL REPRESENTATIONS IN FACTORIZATION FORM

To utilize the previous analyzing framework, we introduce a parameterized representation for modeling the density function of the continuous data. The parameterized model is based on two parts: an early fusion model and an embedding layer which can help us reduce the parameters for classification to finite. An early fusion model can be described as the following. First, $k$ modalities will go through different deep neural networks and output $k$ features. Then, they will be concatenated and fully connected with a $d$-dimensional output layer to learn a joint representation $\boldsymbol{f}$. The embedding layer is the topmost layer for linear classifications, with weights corresponding to label $y$ is given by $\boldsymbol{g}(y) = [g_1(y), \ldots, g_d(y)]^{\mathrm{T}}$. For a specific task, the weights in the topmost layer with a finite number of parameters, i.e. $\boldsymbol{g}(1), \ldots, \boldsymbol{g}(|\mathcal{Y}|)$, can be effectively trained with the joint representation $\boldsymbol{f}$.

Our framework considers an inference model $\tilde{P}_{Y|X_1, \ldots, X_k}^{(\boldsymbol{f}, \boldsymbol{g})}$, which is widely used in natural language processing (Levy & Goldberg, 2014) and image recognition (Xu & Huang, 2020), in the following factorization form[5].

$$\tilde{P}_{Y|X_1, \ldots, X_k}^{(\boldsymbol{f}, \boldsymbol{g})}(y|x_1, \ldots, x_k) \triangleq P_Y(y)(1 + \langle \boldsymbol{f}(x_1, \ldots, x_k), \boldsymbol{g}(y) \rangle). \tag{11}$$

The optimal weights $\boldsymbol{g}_0^*$ and $\boldsymbol{g}_1^*$, which make the model $\tilde{P}_{Y|X_1 \ldots X_k}^{(\boldsymbol{f}, \boldsymbol{g}_i^*)}$ fit the training samples, minimize the distance between empirical distributions and the estimation $P_{X_1 \ldots X_k} \tilde{P}_{Y|X_1 \ldots X_k}^{(\boldsymbol{f}, \boldsymbol{g}_i^*)}$[6], respectively. They can be given as: $\boldsymbol{g}_0^* \triangleq \arg\min_{\boldsymbol{g_0}} \chi_R^2 \left( \hat{P}_{X_1 \ldots X_k Y}, P_{X_1 \ldots X_k} \tilde{P}_{Y|X_1 \ldots X_k}^{(\boldsymbol{f}, \boldsymbol{g}_0)} \right)$, $\boldsymbol{g}_1^* \triangleq \arg\min_{\boldsymbol{g_1}} \chi_R^2 \left( \hat{P}_{X_1 X_2 Y}^{(\mathrm{M})}, P_{XX_1 \ldots X_k} \tilde{P}_{Y|X_1 \ldots X_k}^{(\boldsymbol{f}, \boldsymbol{g}_1)} \right)$, where the fitness is measured by the $\chi^2$-divergence and the reference distribution $R \triangleq P_{X_1 \ldots X_k} P_Y$. This allows us to apply the previous analyses and focus on the inference model $P_{Y|X_1 \ldots X_k}^{(\boldsymbol{f}, \boldsymbol{g}_i^*)}$. Analogous to the linearly combined estimator (2), we consider the linear combination of these inference models

$$Q_{Y|X_1 \ldots X_k}^{(\alpha)} \triangleq (1 - \alpha)\tilde{P}_{Y|X_1 \ldots X_k}^{(\boldsymbol{f}, \boldsymbol{g}_0^*)} + \alpha\tilde{P}_{Y|X_1 \ldots X_k}^{(\boldsymbol{f}, \boldsymbol{g}_1^*)} = \tilde{P}_{Y|X_1 \ldots X_k}^{(\boldsymbol{f}, \boldsymbol{g}^*)}, \tag{12}$$

with $\boldsymbol{g}^* = (1 - \alpha)\boldsymbol{g}_0^* + \alpha\boldsymbol{g}_1^*$.

Further, we define the testing loss and the corresponding optimal coefficient $\alpha^*$ as

$$\tilde{\mathcal{L}}_{\mathrm{test}}^{(\boldsymbol{f}, \boldsymbol{g})}(\alpha) \triangleq \mathbb{E}\left[ \chi_R^2 \left( P_{X_1 \ldots X_k Y}, P_{X_1 \ldots X_k} Q_{Y|X_1 \ldots X_k}^{(\alpha)} \right) \right], \quad \alpha^* \triangleq \underset{\alpha \in [0,1]}{\arg\min} \tilde{\mathcal{L}}_{\mathrm{test}}^{(\boldsymbol{f}, \boldsymbol{g})}(\alpha). \tag{13}$$

We have the following characterization.

---

[5]Note that it can be negative in real applications. But we can also use it to make discriminative decisions through maximum a posterior (MAP) rule.

[6]Note that when the discriminative model $\tilde{P}_{Y|X_1 \ldots X_k}^{(\boldsymbol{f}, \boldsymbol{g})}$ is fixed, $P_{X_1 \ldots X_k} \tilde{P}_{Y|X_1 \ldots X_k}^{(\boldsymbol{f}, \boldsymbol{g})}$ is the optimal approximation of the true distribution $P_{X_1 \ldots X_k Y}$.

**Theorem 5.** *For a given dataset, the optimal $\alpha^*$ for testing loss equation 13 can be given as*
$\alpha^* = \frac{\frac{1}{n}C''}{\Gamma + \frac{1}{n}C'' + \frac{1}{n}V''}$, *where*

$$\Gamma \triangleq \sum_{y \in \mathcal{Y}} \frac{1}{P_Y(y)} \sum_{x_1', \dots, x_k'} \sum_{x_1'', \dots, x_k''} \boldsymbol{f}^{\mathrm{T}}(x_1', \dots, x_k') \Lambda_{\boldsymbol{f}}^{-1} \boldsymbol{f}(x_1'', \dots, x_k'')$$

$$\left( P_{X_1 \dots X_k Y}^{(M)}(x_1', \dots, x_k', y) - P_{X_1 \dots X_k Y}(x_1'', \dots, x_k'', y) \right)^2$$

$$\Lambda_{\boldsymbol{f}} \triangleq \sum_{x_1, \dots, x_k} P_{X_1 \dots X_k}(x_1, \dots, x_k) \boldsymbol{f}(x_1, \dots, x_k) \boldsymbol{f}^{\mathrm{T}}(x_1, \dots, x_k).$$

*Terms $C''$, $V''$, and the calculation approach of those terms from the training data are given in the supplementary material.*

These terms can be represented by some expectations of features $\boldsymbol{f}$ and $\boldsymbol{g}$ and are approximated by the corresponding empirical means. For instance, $\Lambda_{\boldsymbol{f}}$ can be computed from the data by $\Lambda_{\boldsymbol{f}} \leftarrow \frac{1}{n} \sum_{i=1}^{n} \boldsymbol{f}(x_1^{(i)}, \dots, x_k^{(i)}) \boldsymbol{f}^{\mathrm{T}}(x_1^{(i)}, \dots, x_k^{(i)})$.

### 3.2 OUR PROPOSED ALGORITHM

First, based on linear estimator equation 12, the objective function can be chosen as the linear combination of two referenced $\chi^2$-distances measuring the gap between the learned distribution with distributions corresponding to different dependency structures, i.e.,

$$(1 - \alpha)\chi_R^2 \left( \hat{P}_{X_1 \dots X_k Y}, \hat{P}_{X_1 \dots X_k} \hat{P}_{Y|X_1 \dots X_k}^{(\boldsymbol{f}, \boldsymbol{g})} \right) + \alpha \chi_R^2 \left( \hat{P}_{X_1 \dots X_k Y}^{(M)}, \hat{P}_{X_1 \dots X_k} \hat{P}_{Y|X_1 \dots X_k}^{(\boldsymbol{f}, \boldsymbol{g})} \right). \quad (14)$$

Based on the approach established in (Wang et al., 2019; Huang et al., 2017), the objective equation 14 can then be transformed to the following loss function which can be computed by the feature $(\boldsymbol{f}, \boldsymbol{g})$,

$$\tilde{\mathcal{L}}_{\mathrm{train}}^{(\alpha)}(\boldsymbol{f}, \boldsymbol{g}) = (1 - \alpha)\mathcal{L}_{dep}(\boldsymbol{f}, \boldsymbol{g}) + \alpha \mathcal{L}_{dep}^{(M)}(\boldsymbol{f}, \boldsymbol{g}), \quad (15)$$

$$\mathcal{L}_{dep}(\boldsymbol{f}, \boldsymbol{g}) = \frac{1}{n-1} \sum_{i=1}^{n} \boldsymbol{f}^{\mathrm{T}}(x_1^{(i)}, \dots, x_k^{(i)}) \boldsymbol{g}(y^{(i)}) - \frac{1}{2} \mathrm{tr}(\mathrm{cov}(\boldsymbol{f}) \mathrm{cov}(\boldsymbol{g})), \quad (16)$$

$$\mathcal{L}_{dep}^{(M)}(\boldsymbol{f}, \boldsymbol{g}) = \sum_{j=1}^{m} \hat{P}_Y(j) \left[ \frac{1}{n_j - 1} \sum_{i=1}^{n_j} \boldsymbol{f}^{\mathrm{T}}(\underline{x}_1^{(i,j)}, \dots, \underline{x}_k^{(i,j)}) \boldsymbol{g}(j) - \frac{1}{2} \mathrm{tr}(\mathrm{cov}(\boldsymbol{f}_j) \mathrm{cov}(\boldsymbol{g})) \right], \quad (17)$$

where $\boldsymbol{f}(x_1^{(i)}, \dots, x_k^{(i)})$ is the feature output for $i$-th sample $(x_1^{(i)}, \dots, x_k^{(i)}, y^{(i)})$, $\boldsymbol{g}(i)$ is the embedding for label $i$, $\mathrm{cov}(\boldsymbol{f}) \leftarrow \frac{1}{n-1} \sum_{i=1}^{n} \boldsymbol{f}(x_1^{(i)}, \dots, x_k^{(i)}) \boldsymbol{f}^{\mathrm{T}}(x_1^{(i)}, \dots, x_k^{(i)})$, $\mathrm{cov}(\boldsymbol{g}) \leftarrow \frac{1}{n-1} \sum_{i=1}^{n} \boldsymbol{g}(y^{(i)}) \boldsymbol{g}^{\mathrm{T}}(y^{(i)})$, $\hat{P}_Y(j) = \sum_{i=1}^{n} \mathbb{1}\{y^{(i)} = j\}, i = 1, \dots, m$. As for loss $\mathcal{L}_{dep}^{(M)}(\boldsymbol{f}, \boldsymbol{g})$, it needs a permutation on samples' modalities within the subset of the same label. We denote the subset of training samples with label $j \in \{1, \dots, k\}$ as $\mathcal{D}_j = \{(x_1^{(i,j)}, \dots, x_k^{(i,j)})\}_{i=1}^{d_j}$, where $d_j$ is the number of samples whose label is $j$ in the overall dataset $\mathcal{D}$. $\underline{x}_t^{(i,j)}$ is chosen from $\{x_t^{(i,j)}\}_{i=1}^{d_j}, t = 1, \dots, k$, and $n_j = \prod_{t=1}^{k} d_t$, $\mathrm{cov}(\boldsymbol{f}_j) \leftarrow \frac{1}{n_j - 1} \sum_{t=1}^{n_j} \boldsymbol{f}(x_1^{(t,j)}, \dots, x_k^{(t,j)}) \boldsymbol{f}^{\mathrm{T}}(x_1^{(t,j)}, \dots, x_k^{(t,j)})$.

Then, our algorithm can be organized as an iteration of two main optimizations: (i) the optimization of $\alpha$ for given $(\boldsymbol{f}, \boldsymbol{g})$ by minimizing the testing loss $\tilde{\mathcal{L}}_{\mathrm{test}}(\alpha)$ equation 13 and use the expression in Theorem 5 for computation; (ii) the optimization of features $(\boldsymbol{f}, \boldsymbol{g})$ for given $\alpha$ to minimize the training loss equation 14 by the deep neural network. We summarize it in Algorithm 1.

With the output features $\boldsymbol{f}^*$ and $\boldsymbol{g}^*$ trained by the algorithm, the classification of a newly observed sample $(x_1, x_2)$ is given by the maximum a posterior (MAP) decision rule

$$\tilde{y}(x_1, \dots, x_k) = \arg\max_{y \in \mathcal{Y}} P_{Y|X_1 \dots X_k}(y|x_1, \dots, x_k)$$

$$= \arg\max_{y \in \mathcal{Y}} P_Y(y)(1 + \langle \boldsymbol{f}^*(x_1, \dots, x_k), \boldsymbol{g}^*(y) \rangle).$$

---

**Algorithm 1** An Auto-updated Coefficient on Dependency Structures (auto-CODES) Algorithm

---

**Input:** multimodal data samples $\{(x_1^{(i)}, \ldots, x_2^{(i)}, y^{(i)})\}_{i=1}^n$
Initialize $\alpha^* = 0$
**repeat**
$\quad (\boldsymbol{f}^*, \boldsymbol{g}^*) \leftarrow \arg\min_{\boldsymbol{f}, \boldsymbol{g}} \tilde{\mathcal{L}}_{\text{train}}^{(\alpha^*)}(\boldsymbol{f}, \boldsymbol{g})$
$\quad \alpha^* \leftarrow \arg\min_{\alpha \in [0,1]} \tilde{\mathcal{L}}_{\text{test}}^{(\boldsymbol{f}^*, \boldsymbol{g}^*)}(\alpha)$
**until** $\alpha^*$ converges
$(\boldsymbol{f}^*, \boldsymbol{g}^*) \leftarrow \tilde{\mathcal{L}}_{\text{train}}^{(\alpha^*)}(\boldsymbol{f}, \boldsymbol{g})$
**return** $\boldsymbol{f}^*, \boldsymbol{g}^*, \alpha^*$

---

## 4 EXPERIMENTS

In this section, we verify our model and algorithm to answer the following research questions (RQ):

**RQ1:** *Can auto-CODES make discrimination well?*

**RQ2:** *Can auto-CODES automatically determine an appropriate coefficient $\alpha$?*

**RQ3:** *Is the optimal coefficient $\alpha$ scale to the inverse proportion of the training sample size?*

**Experimental settings.** In our experiments, two widely-used multimodal emotion recognition datasets are used, MELD (Poria et al., 2018) and IEMOCAP (Busso et al., 2008). MELD contains 13K utterances from 1433 dialogues from the TV series *Friends*. Each utterance is annotated with three emotion labels, positive, neutral, and negative. We use the audio and textual modalities for our verification. As for IEMOCAP, it contains approximately 12 hours of audiovisual data with six emotion categories: anger, happiness, sadness, neutral, excitement, and frustration. We use visual and audio modalities for our verification. In our settings, the sample size plays a crucial role. Thus, we randomly select subsets of both datasets with certain levels of sample sizes as our training sets. To preserve the inner structure within dialogues, the random selection is towards the dialogues. The exact sample sizes are listed in Table 1. As for the model structure, we use DialogueRNN (Majumder et al., 2019) as our backbone network for extracting multi-modal features and adopt 2-layer Multi-layer Perceptron (MLP) with RELU (Glorot et al., 2011) activation for extracting one-hot label features. We use accuracy and F1-score as our evaluation metric.

### 4.1 EMOTION RECOGNITION RESULTS

To answer RQ1, we compare auto-CODES with the following methods: (i) CE: Cross entropy loss that is widely used in machine learning classification tasks, (ii) MaskedNLL: a variant of NLL loss to cope with excluding logit value of the padded sample, which is used in (Majumder et al., 2019), (iii) Focal loss(Mukhoti et al., 2020): Focal loss is designed to address the issue of the class imbalance problem, and (iv) Soft-HGR (Wang et al., 2019): Soft-HGR loss learns correlated representation across modalities without hard whitening constraints. All the experiments are conducted on various training sample sizes. The discrimination accuracies along with the F1-score on emotion recognition tasks are reported and shown in Table 1 and Table 3.

The results demonstrate that our auto-updated method achieves superior performance against existing methods among all settings with different sample sizes. From the dialogue size 10 to 40, auto-CODES outperforms the second-best method Soft-HGR by the margin of 2.086% (size 20) to 2.63% (size 30) on the F1-score metric. For the dialogue sizes 40 and 60, auto-CODES achieves absolute improvements by margins of 1.629% and 1.515% over Focal loss. Also, to examine the impact of the coefficient $\alpha$, we conduct experiments with prefixed static $\alpha$ as a comparison with our auto-updated algorithm. Comparing the last two columns in Table 1, we can observe that auto-CODES obtains 1.856% (size 60) to 4.456% (size 10) improvements in terms of F1-score over static CODES. These results suggest that: (1) when the training samples are insufficient, our proposed auto-CODES outperforms focal loss, Soft-HGR, Cross Entropy loss, and MaskedNll loss, (2) auto-updating $\alpha$ can improve the results over static CODES by large margins.

Table 1: Comparison with other objectives in MELD dataset with different training sample sizes. All reported results are averaged over 10 repeated experiments.

| Sample size (Dialogue size) | Metric | Method | | | | | |
|---|---|---|---|---|---|---|---|
| | | CE | MaskedNll | Focal | Soft-HGR | CODES$_{\alpha=0.05}$ | auto-CODES |
| **107 (10)** | accuracy | 50.222±2.709 | 50.544±1.380 | 50.858±3.005 | 52.784±2.143 | 51.741±3.221 | **53.640±1.345** |
| | F1-score | 38.099±2.296 | 40.147±2.269 | 40.649±1.999 | 44.998±2.227 | 42.751±1.683 | **47.206±1.179** |
| **198 (20)** | accuracy | 51.218±1.312 | 51.571±1.255 | 51.226±1.477 | 53.890±1.879 | 53.091±2.404 | **55.179±1.147** |
| | F1-score | 41.840±1.657 | 42.912±1.615 | 42.535±2.146 | 47.038±1.957 | 45.786±1.210 | **49.124±0.902** |
| **302 (30)** | accuracy | 51.272±1.467 | 52.084±0.990 | 52.655±0.421 | 54.736±1.576 | 53.986±1.407 | **55.943±0.925** |
| | F1-score | 42.091±1.275 | 43.351±1.527 | 43.589±1.986 | 48.358±1.170 | 46.816±2.313 | **50.988±0.712** |
| **413 (40)** | accuracy | 52.475±0.979 | 52.943±1.159 | 53.471±0.534 | 55.441±1.362 | 54.219±1.513 | **57.816±0.964** |
| | F1-score | 43.647±0.764 | 44.341±1.165 | 46.135±1.146 | 49.213±0.927 | 47.942±1.252 | **51.538±0.699** |
| **516 (50)** | accuracy | 54.176±1.098 | 53.586±1.042 | 55.613±0.665 | 55.901±0.937 | 54.730±1.467 | **58.467±0.783** |
| | F1-score | 47.397±1.100 | 46.537±0.934 | 50.509±1.700 | 50.107±0.776 | 48.523±1.012 | **52.138±0.872** |
| **636 (60)** | accuracy | 55.448±0.707 | 55.441±0.941 | 56.576±0.479 | 56.467±0.792 | 56.087±1.280 | **59.847±0.781** |
| | F1-score | 50.547±0.876 | 50.263±1.231 | 51.138±0.944 | 50.981±0.811 | 50.747±0.812 | **52.653±0.584** |

Table 2: The optimal coefficients $\alpha^*$ derived by auto-CODES and grid search method on different training sample sizes on the MELD dataset.

| Sample size n (dialogue size) | grid search (gs) | | auto-CODES (auto) | | |
|---|---|---|---|---|---|
| | $\alpha_{gs}$ | accuracy | $\alpha_{auto}$ | accuracy | $n \cdot \alpha_{auto}$ |
| 107 (10) | 0.02 | 52.912 ± 1.164 | 0.0189 ± 0.0011 | 53.640 ± 1.345 | 3.74 |
| 198 (20) | 0.02 | 55.144 ± 1.470 | 0.0179 ± 0.0009 | 55.179 ± 1.147 | 3.54 |
| 302 (30) | 0.01 | 55.949 ± 1.342 | 0.0130 ± 0.0016 | 55.943 ± 0.925 | 3.93 |
| 413 (40) | 0.01 | 57.486 ± 1.292 | 0.0092 ± 0.0007 | 57.816 ± 0.964 | 3.81 |
| 516 (50) | 0.01 | 58.372 ± 0.964 | 0.0078 ± 0.0008 | 58.467 ± 0.783 | 4.02 |
| 636 (60) | 0.01 | 59.573 ± 0.745 | 0.0058 ± 0.0006 | 59.848 ± 0.781 | 3.69 |

## 4.2 DETERMINATION OF THE OPTIMAL COEFFICIENT

To answer RQ2, we conduct experiments to compare our auto-updated coefficient $\alpha$ and the coefficient determined by grid search and manual adjustments conventionally. Instead of only relying on empirical experience with no theoretical assurance, our theory determines the optimal coefficient $\alpha^*$ and updates the learned modalities' features automatically.

During training, our algorithm and grid search method use the same number of training epochs for each $\alpha$ updating iteration circle at different training tuple sizes. In the grid search method, we search 101 values of $\alpha$ ranging from 0 to 1 with a step length of 0.01. In our algorithm, we stop the iteration once the difference ratio of the updating alpha is smaller than 0.1. The results are shown in Table 2. It suggests that our method can locate an appropriate and even better $\alpha$ than the grid search method.

## 4.3 OPTIMAL COEFFICIENT WITH THE NUMBER OF TRAINING SAMPLES

To answer RQ3, we examine the optimal $\alpha$ determined by auto-CODES on MELD and IEMOCAP. According to our theory, $\alpha^*$ is roughly of the order $\frac{1}{n}$. For different training sample sizes, we conduct 10 repetitive experiments using auto-CODES to locate the optimal coefficient and report their average as the final $\alpha^*$. The values of their product $n \cdot \alpha^*$ for both MELD dataset are demonstrated in Table 2. It can be approximately recognized as a fixed number around 3.8. The results for IEMOCAP has given in the appendix. According to the column "$n \cdot \alpha_{auto}$" of Table 2, our theoretical outcome can not only determine a good $\alpha$ but also reveal its relation with the number of training sample sizes. For more experiment results on the IEMOCAP dataset, please see Supplementary Material.

Table 3: Comparison with other objectives in the IEMOCAP dataset with different training sample sizes. All reported results are averaged over 10 repeated experiments.

| Sample size (Dialogue size) | Metric | Method | | | | | |
|---|---|---|---|---|---|---|---|
| | | CE | MaskedNll | Focal | Soft-HGR | CODES$_{\alpha=0.01}$ | auto-CODES |
| **446 (10)** | accuracy | 42.580 ± 1.791 | 41.956 ± 1.645 | 42.890 ± 1.783 | 42.466 ± 1.980 | 42.784 ± 1.465 | **43.023 ± 1.706** |
| | F1-score | 36.710 ± 2.009 | 35.518 ± 1.480 | 37.302 ± 1.592 | 36.398 ± 1.848 | 37.010 ± 1.573 | **37.415 ± 1.292** |
| **565 (12)** | accuracy | 43.217 ± 1.487 | 44.789 ± 1.669 | 44.962 ± 1.374 | 44.511 ± 1.655 | 44.754 ± 1.221 | **45.478 ± 1.236** |
| | F1-score | 37.399 ± 1.642 | 40.726 ± 0.878 | 40.519 ± 1.587 | 39.507 ± 1.132 | 40.751 ± 1.383 | **40.910 ± 1.085** |
| **647 (14)** | accuracy | 45.116 ± 1.054 | 45.533 ± 1.577 | 45.599 ± 1.568 | 44.969 ± 1.513 | 45.730 ± 1.420 | **46.239 ± 1.128** |
| | F1-score | 40.714 ± 1.199 | 41.227 ± 1.114 | 41.376 ± 1.247 | 40.397 ± 1.022 | 41.786 ± 1.209 | **42.803 ± 0.976** |

## 5 RELATED WORK

In this section, we summarize the related works from two aspects: correlation analysis and the dependency structures in multimodal learning.

**Correlation Analyses in Multimodal Learning.** Correlation analysis methods can be fusion-based or objective-based. Fusion-based methods design the fusion approaches of different modalities representations, such as tensor fusion network for multimodal sentiment analysis (Zadeh et al., 2017), and factorized multimodal representations (Liu et al., 2018; Tsai et al., 2018). Objective-based methods capture modality interactions by using distinct statistical notions. For example, canonical correlation analysis (CCA) approaches like (Karami & Schuurmans, 2021) are based on Pearson correlation. Jensen-Shannon-Divergence (Sutter et al., 2020), Variation of Information (Sohn et al., 2014), and HGR maximal correlation (Wang et al., 2019; Ma et al., 2020; Tong et al., 2021) have also shown that the statistical design of learning objectives can facilitate the correlation extraction among modalities. The concept of sample efficiency has been discussed in the weighted algorithm TAWT (Chen et al., 2021). The explicit and accurate quantity of training samples for analysis is not appropriately addressed.

**Learning Dependency Structures in Multimodal.** Learning features in complex manifolds formed by different modalities of data is essential. One approach is to learn the discriminative one such as conditional random fields (CRF) (Lafferty et al., 2001). Another strategy involves learning the generative model using multimodal deep Boltzmann machines (DBMs) (Salakhutdinov & Hinton, 2009), or coupled, factorial and multi-stream hidden Markov models (HMM) method (Nefian et al., 2002; Ghahramani & Jordan, 1997; Gurban et al., 2008). However, the theoretical characterization of utilizing the dependency structures is not sufficiently explored.

## 6 DISCUSSION

In this paper, we propose a new theoretical framework to analytically characterize the explicit and exact relation between the sample size with conditional dependency structures in multimodal learning in a non-asymptotic regime. Moreover, we propose a weighted training algorithm, auto-CODES, based on the theoretical framework. It can iteratively update the coefficient on different dependency structures based on the evolving modalities' features. The effectiveness of auto-CODES is further corroborated through multimodal emotion recognition experiments on MELD and IEMOCAP datasets with promising results.

**Limitations and Future Work.** There are two main limitations in our work. First, we proposed a tractable algorithm for one specific type of conditional dependency structure, but we left the generalized tractable approach for all different kinds of dependency structures for future work. Second, even though we have specified the precise shape of the coefficient that features can compute, it is still laborious to specify its role in the algorithm; therefore, we intend to improve the computing method for the coefficient in the future. We also intend to integrate our framework with pre-train networks and run further experiments on various modalities across various datasets, including MOSEI.

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
