# OpenReview forum: "A Mathematical Framework for Characterizing Dependency Structures of Multimodal Learning"
_ICLR.cc/2023/Conference — Submitted to ICLR 2023_

### Official Review · Reviewer_MdJu · 2022-10-21

**Confidence:** 3
**Clarity, Quality, Novelty And Reproducibility:** This paper writes clearly and propose…
**Correctness:** 3
**Technical Novelty And Significance:** 3
**Empirical Novelty And Significance:** 3
**Recommendation:** 6

**Strength And Weaknesses:**

Strength:
+ This paper writes well and explains the formulations and analytical results clearly.
+ The authors first analyze the discrete scenarios and conclude some theoretical understanding for model designing with the formulation of optimal combination coefficient, then extend to the continuous scenarios to propose an autonomous updated coefficient on dependency structures algorithm by exploiting parametric models.
+ The experiments demonstrate the effectiveness of this algorithm.

Weaknesses:
- This work suggests models learn the linear combination of general dependency structure and conditional dependency structure estimators, which is a strong assumption, especially in multimodal scenarios, for example, in audio-visual scenarios, whatever the label is, it usually contains high temporal consistency between vision and sound.
- The experiments only provide small-sale results, it cannot prove the conclusion useful for the large-scale dataset.
- This paper point out three factors affecting the multimodal classification performance, but the results only demonstrate the results with different training samples sizes, however, in different datasets, the intrinsic sample structures are various, and the estimated $\chi ^2-divergence$ in different datasets should also be provided to prove the algorithm.
- The coefficient is estimated with training samples, however, the name of loss to estimate is $L_{test}$, which may be ambiguous.

**Summary Of The Paper:**

This paper proposes a mathematical framework to characterize the conditional dependency structures for multimodal learning in an analytic way. Concretely, the authors consider three factors affecting the multimodal classification performance including the number of labeled training samples, the fitness of the conditional dependency structures to represent the true joint distribution, and the complexity of the discrimination task, where the $\chi ^2-divergence$ is utilized to estimate the fitness of the conditional dependency structures.
The authors consider a model to learn the linear combination of the general dependency structure and conditional dependency structure estimators and propose an explicit analytical solution to get the optimal combination coefficient.
The authors not only show some theoretical analysis for designing the efficient multimodal algorithm but also provide experiment results to show the effectiveness of this algorithm.

**Summary Of The Review:**

This paper proposes a novel mathematical framework to characterize the conditional dependency structures for multimodal learning in an analytic way and provides experiments to prove the theory and algorithm. However, it still needs more solid results to prove the results.

---

### Official Review · Reviewer_pfqE · 2022-10-24

**Confidence:** 4
**Correctness:** 3
**Technical Novelty And Significance:** 2
**Empirical Novelty And Significance:** 2
**Recommendation:** 1

**Clarity, Quality, Novelty And Reproducibility:**

The writing of this paper is very poor, which let me not able to understand the core idea of this paper.

**Strength And Weaknesses:**

1.  The writing of this paper is very poor. There are too many typos:
- 'Learning' is missing in the title;
- 'from different categories of data, such as images, audios, and texts' 'categories' should be substituted to 'types';
- 'There have been many kinds of literature addressing this issue with different kinds of algorithms', but only one reference is cited here;
- 'training sample size' $\rightarrow$ 'training dataset size'
2. Some concepts are not clear:
- 'design feature' and 'extract feature' are both mentioned in the introduction, but these are two different concepts;
- ' exploiting appropriate dependency structures' and 'representing the general dependency structure' are both mentioned in the introduction. So the dependency structure can be straightly adopted or need to learn and capture?
3.  No enough experiments to support the proposed method.




**Summary Of The Paper:**

This paper focuses on multi-modal learning problem, and proposes a mathematical framework to characterize so-called dependency structures.

**Summary Of The Review:**

The writing of this paper is very poor, which let me not able to understand the core idea of this paper.

---

### Official Review · Reviewer_kPWv · 2022-10-25

**Confidence:** 2
**Clarity, Quality, Novelty And Reproducibility:** The code is provided and the analysis…
**Correctness:** 3
**Technical Novelty And Significance:** 3
**Empirical Novelty And Significance:** Not applicable
**Recommendation:** 5

**Strength And Weaknesses:**

Strengths: The authors did a very comprehensive theoretical analysis, and the proof looks sound.

Weaknesses: The paper is not easy to follow and read.



**Summary Of The Paper:**

This paper proposes a theoretical framework to study the relationship between sample size with conditional dependency structures in multimodal learning. The method's effectiveness is validated on two emotion recognition experiments.

**Summary Of The Review:**

Overall, I think the paper did a great work on the theoretical analysis. However, the weakness is the paper's readability.

---

### Official Review · Reviewer_c4xb · 2022-10-25

**Confidence:** 3
**Correctness:** 3
**Technical Novelty And Significance:** 3
**Empirical Novelty And Significance:** 3
**Recommendation:** 5

**Clarity, Quality, Novelty And Reproducibility:**

The introduction is hard to follow, which compromises the motivation of the proposed work.
Essential background information is missing, which obscures the justification of the contribution of the proposed work.

**Strength And Weaknesses:**

Strength :
+ A theoretical framework is proposed to characterize the optimal combination coefficients between the empirical joint distribution and Markov-structured distribution.

+ The experimental designs are clear and straightforward, with solid results to verify the proposed algorithm.


Weaknesses:
- The proposed research is not well motivated. The introduction section is hard to follow in general. For instance, the introduction starts with the issue of insufficient training samples in multimodal learning. It is unclear how this can logically motivate the proposed optimal combination coefficients study.

- Missing key background information. Eq 2 defines the combination coefficients alpha. However, it is completely unclear why the two distributions are selected and why they can be linearly combined using alpha. This makes the problem definition ungrounded and hard to follow.

- Unclear key technical details. When calculating referenced x^2-divergence, how to obtain the ground-truth distribution P?

- Experimental issues. All experiments are limited to two modalities. It is unclear if the proposed algorithm can handle more than two modalities.
(Table 2) For the first row (sample size 107), please check the result of  “n・\alpha.”


**Summary Of The Paper:**

This paper studied the dependency structure between modalities in multimodal learning. Specifically, it intends to understand the interplay between training samples and various dependency structures. A novel theoretical framework is proposed to characterize the training sample size toward different dependency structures.


**Summary Of The Review:**

The paper needs further improvements to justify the motivation and clearly specify its theoretical and technical contributions.

---

### Decision · Program_Chairs · 2023-01-20

**Decision:**

Reject

**Justification For Why Not Higher Score:**

The practical significance of the modeling assumption is unclear. In fact the paper is not connected with existing literature and the reviewers generally find it hard to read.

**Justification For Why Not Lower Score:**

N/A

**Metareview: Summary, Strengths And Weaknesses:**

The paper propose to model dependency structures of multi-modal learning with a linear combination of two distributions, one is the un-structured joint distribution (over view 1, view 2, and label), while the one has a conditional independence structure (conditioned on label, input views are independent).

Strength:
The authors derived finite sample analysis for the estimator, and proposed algorithms for estimating the combination coefficient and the predictors.

Weakness:
The paper is largely detached with the multi-modal learning literature. The strategy of modeling multi-modal with the mixture of the two distributions seem to suggest some data samples are not structured whereas others are; this choice is non-intuitive (we often assume there is certain dependency structure that all data samples follow).  While we appreciate the authors' theoretical analysis on their estimator, the intuition is not well justified. The experiments do not compare the proposed method with alternative methods and different modeling assumptions (and merely compare different supervised losses).